# Detectionof Major Depressive Disorder Based on a Combination of Voice Features: An Exploratory Approach

**DOI:** 10.3390/ijerph191811397

**Published:** 2022-09-10

**Authors:** Masakazu Higuchi, Mitsuteru Nakamura, Shuji Shinohara, Yasuhiro Omiya, Takeshi Takano, Daisuke Mizuguchi, Noriaki Sonota, Hiroyuki Toda, Taku Saito, Mirai So, Eiji Takayama, Hiroo Terashi, Shunji Mitsuyoshi, Shinichi Tokuno

**Affiliations:** 1Department of Bioengineering, Graduate School of Engineering, The University of Tokyo, Tokyo 113-8656, Japan or; 2School of Science and Engineering, Tokyo Denki University, Saitama 350-0394, Japan; 3PST Inc., Yokohama 231-0023, Japan; 4Department of Psychiatry, School of Medicine, National Defense Medical College, Saitama 359-8513, Japan; 5Department of Neuropsychiatry, Tokyo Dental College, Tokyo 101-0061, Japan; 6Department of Oral Biochemistry, Asahi University School of Dentistry, Gifu 501-0296, Japan; 7Department of Neurology, Tokyo Medical University, Tokyo 160-8402, Japan; 8Graduate School of Health Innovation, Kanagawa University of Human Services, Yokosuka 210-0821, Japan

**Keywords:** voice analysis, major depressive disorder, logistic regression

## Abstract

In general, it is common knowledge that people’s feelings are reflected in their voice and facial expressions. This research work focuses on developing techniques for diagnosing depression based on acoustic properties of the voice. In this study, we developed a composite index of vocal acoustic properties that can be used for depression detection. Voice recordings were collected from patients undergoing outpatient treatment for major depressive disorder at a hospital or clinic following a physician’s diagnosis. Numerous features were extracted from the collected audio data using openSMILE software. Furthermore, qualitatively similar features were combined using principal component analysis. The resulting components were incorporated as parameters in a logistic regression based classifier, which achieved a diagnostic accuracy of ~90% on the training set and ~80% on the test set. Lastly, the proposed metric could serve as a new measure for evaluation of major depressive disorder.

## 1. Introduction

The importance of mental health care in managing different types of stress in modern society is increasingly recognized around the world. Stress has negative effects on people’s health and mood in daily life, and its accumulation can cause mental and behavioral dysfunction in the long term [1]. Besides their impact on individuals, such disorders result in serious economic costs to society because of their association with reduction of lifetime earnings and labor productivity [2,3]. This state of affairs requires technologies that are capable of easily checking for mental illnesses such as depression, for which early intervention is associated with higher remission rates [4,5,6].

Previously, some researchers focused on identifying biomarkers in saliva and blood for use in depression screening [7,8,9]. For example, Maes et al. proposed interleukin-1 receptor antagonist (IL-1ra) as a diagnostic biomarker for major depressive disorder (MDD), finding its serum concentration to be increased in affected individuals [9]. However, besides being invasive, diagnostic body fluid testing incurs additional costs because of the need for special measurement equipment and reagents. Self-report psychological questionnaires such as the Patient Health Questionnaire 9 (PHQ9), General Health Questionnaire (GHQ), and Beck Depression Inventory (BDI) are non-invasive alternatives commonly used by clinicians [10,11,12]. They are relatively simple to administer, but suffer from the inherent drawback of reporting bias, i.e., certain symptoms/behaviors being over- and/or under-endorsed depending on respondents’ awareness of them (or lack thereof) [13]. This bias can be mitigated by assessments conducted by physicians, such as the Hamilton Depression Rating Scale (HDRS); however, the extra time involved limits the number of interviews that can be administered [14].

Feelings are reflected in people’s voice and facial expressions, and this common knowledge has also been scientifically substantiated [15,16]. Such evidence has driven a recent surge of research interest in acoustic biomarkers for predicting depression and stress levels [17,18,19]. Simplicity is a major advantage of such approaches, i.e., voice recordings can be collected non-invasively and remotely without any specialized equipment besides a microphone. Furthermore, they reduce subjectivity in diagnosis since recording data are processed algorithmically; thus, they avoid reporting bias that is inherent to self-report assessments, holding promise for detecting a variety of mental illnesses. For example, Mundt et al. recorded MDD patients reading a standard script via a telephone interface, and calculated a selection of vocal acoustic properties such as the duration and ratio of vocalizations and silences, fundamental frequency (F0), and first and second formants (F1, F2). Several of these measures were markedly different between patients who responded to treatment and those who did not [18]. Our research group has also focused on depression’s association with emotional expression. In a previous study, we developed composite metrics for quantifying mental health—“vitality” and “mental activity”—that combine different emotional components of the voice [19]. In subsequent work, we showed evidence for this measure’s effectiveness in detecting depression and monitoring stress due to life events [20,21]. A weak correlation between vitality and BDI score was confirmed, suggesting that some voice features correlated with the BDI score. Still, some limitations of this measure have become apparent. First, since diseases besides depression affect how emotions are expressed, it is challenging to resolve whether abnormal vitality and mental activity truly indicates depression or instead reflects a different condition. Furthermore, since its classification accuracy showed large variation across facilities in some cases, vitality and mental activity might be dependent on (recording) environment.

openSMILE is a platform for deriving extensive sets of acoustic features from audio data [22], which has been recently applied by several studies in the field of speech diagnostics. Jiang et al. developed a novel computational methodology for detecting depression based on vocal acoustic features extracted using openSMILE from recorded speech in three categories of emotion (positive, neutral, negative). Despite obtaining high accuracy for depression detection, the development of separate models for men and women slightly complicated their application in practice [23]. Faurholt-Jepsen et al. extracted openSMILE features from voice recordings of patients with bipolar disorder, and attempted to use them to classify their depressive and manic symptoms. Their feature-based classification accurately matched manic and depressive symptoms as measured by the Young Mania Rating Scale (YMRS [24]) and HRDS, respectively. Nevertheless, their algorithm utilizes an immense number of features, posing a risk of overfitting [25]. Focusing on mel-frequency cepstrum coefficients (MFCCs), Taguchi et al. reported a significant difference in the second coefficient (MFCC2), which represents spectral energy in the 2000–3000 Hz band, between the voices of MDD patients and healthy controls. However, their analysis included only one type of feature, and did not combine multiple features [26]. In a previous study, we proposed a voice index based on openSMILE features, which could accurately differentiate between three subject groups: patients with major depressive disorder, patients with bipolar disorder, and healthy individuals. Still, the proposed measure requires further validation because our training data were drawn from a small sample [27].

The aim of this paper is to develop a composite index based on vocal acoustic features that can accurately differentiate patients with major depressive disorder from healthy adults. Depressed patients and non-depressed controls were recorded reading a set of fixed phrases; the recorded data were split into training and test datasets. Features were extracted from the training data using openSMILE, and qualitatively similar features were mathematically aggregated by means of principal component analysis. These components were used as coefficients in logistic regression to classify subjects. The classification performance of our proposed indicator was tested on the recordings of the test dataset. The result achieved a diagnostic accuracy of approximately 80%.

## 2. Materials and Methods

### 2.1. Ethical Considerations

This study was approved by the institutional review board of The University of Tokyo (no. 11572).

### 2.2. Subjects

Our study enrolled 306 subjects from five institutions in total. Depressed subjects were recruited from individuals receiving outpatient treatment for major depressive disorder at Ginza Taimei Clinic (“C”: 87) or National Defense Medical College Hospital (“H1”: 90). For the control group, self-reported healthy adults were recruited from the National Defense Medical College Hospital (14), Tokyo Medical University Hospital (“H2”: 23), Asahi University (“U1”: 38), and The University of Tokyo (“U2”: 54). Patients gave informed consent after receiving the study information at their first assessment. Controls gave informed consent after receiving the study information at a health workshop held by the authors, either individually or in small groups. Patients aged 20 years or older were enrolled if they met the diagnostic and statistical manual of mental disorders 4th edition Text Revision (DSM-IV-TR) diagnostic criteria for major depressive disorder [28]. Candidates with severe physical disability or organic brain disease were excluded. The subjects were diagnosed by a psychiatrist using the Mini-International Neuropsychiatric Interview (M.I.N.I.) [29]. The details for the subjects recruited from each facility are summarized in Table 1.

The severity of depression was assessed by physicians using the HDRS. Several standards for HDRS severity rating have been reported [30]; following the precedent of Riedel et al. [31], we interpreted that a HDRS total score of less than 8 indicates depression in remission and excluded patients accordingly from analysis. Our final study population consisted of 102 patients (“depressed group (HDRS ≥ 8)”) and 129 healthy adults (“normal group”). The screening outcomes of patients recruited for the “depressed group” are summarized in Table 2.

### 2.3. Voice Data

Voice recordings were made in an examination room (C, H1, H2) or conference room (U1, U2). Each subject read aloud 10 set phrases in Japanese, given along with their English translations in Table 3. Voice recordings were acquired at 24 bit and 96 kHz resolution using a Roland R-26 portable digital audio recorder (Hamamatsu, Japan) and Olympus ME52W lavalier microphone (Tokyo, Japan).

### 2.4. Voice Analysis

Each audio file was first normalized to minimize differences in volume due to recording environment, and then segmented by phrase. Each phrase was processed independently to extract vocal features. Various scripts were available for automatically calculating different sets of features from audio data using openSMILE. Our study used “the large openSMILE emotion feature set”, developed for use in emotion recognition. We used 6552 (56 × 3 × 39) audio features computed as follows:56 types of acoustic/physical quantities were calculated at frame level as low-level descriptors: fast Fourier transform (FFT), MFCCs, voiced speech probability, zero-crossing rate, signal energy, F0, and so on.3 types of temporal statistics were derived from these descriptors at frame level: moving average, first-order change over time (“delta”), and second-order change over time (“delta-delta”).39 types of statistical functionals were calculated at file (phrase) level from frame-level values: mean, maximum, minimum, centroid, quartiles, variance, kurtosis, skewness, and so on.

Each feature was averaged for every subject across the 10 set phrases to obtain mean values for analysis. The processed data were split by facility of origin into a training set (C, H2, U1, U2) and test set (H1). Next, the classification algorithm was trained on the training data using the derived features.

Since models tend to overfit if trained on too many variables, we reduced dimensionality through a combination of receiver operating characteristic (ROC) analysis and principal component analysis (PCA). First, each feature’s ability to independently distinguish depressed from normal adults was quantified as the area under the corresponding ROC curve (AUC); only features exceeding a certain threshold were selected. Next, highly correlated features were transformed into principal components. We carefully selected fewer features than our sample size given that PCA cannot be applied when more features than subjects are present because the resulting correlation matrix is rank-deficient. These components’ ability to predict depression was tested by logistic regression with L2 regularization (ridge regression) [32]. The regularization parameters were optimized by cross-validation within the training set. Since data were split randomly during cross-validation, different values were computed for the regularization parameters every time the algorithm was run, causing downstream variation in regression coefficients. To stabilize the results of training, the model was trained several times, and each coefficient was averaged across runs to obtain the final weights.

The classification model was a logistic function describing the linear sum of three PCs (parameters) times their respective coefficients (weights). This function’s output was adopted as the classification index for major depressive disorder. The diagnostic performance of this metric was tested on the voice recordings in the test dataset (H1).

To determine the training and test data sets, splitting the data independent of the institution is also possible. However, to facilitate future applications, we employed a method where only a part of the recording environment data, as opposed to the whole, is incorporated in the index. This allows us to assess the extent to which voices in other recording environments were not incorporated in the index. Since H1 comprised of both healthy and depressed subjects, we used the speech data of H1 subjects as the test data.

Statistical processing was conducted using the free software R (version 4.0.2) [33].

## 3. Results

### 3.1. Feature Selection

openSMILE features whose AUC exceeded our threshold (≥0.869) were selected. Our model incorporated 187 features, fewer than the number of recordings in the training dataset (n=192).

### 3.2. Principal Component Analysis

Through PCA, three components were extracted from the 187 features of the training data, cumulatively accounting for 80% of the observed variance.

### 3.3. Logistic Regression with Regularization

Logistic regression with L2 regularization using the three components as predictors was performed 20 times. Table 4 presents the obtained mean regression coefficients.

Our novel indicator—Major Depression Discrimination Index (MDDI)—is given by the following formula: (1)$$\mathrm{MDDI}=\frac{1}{1+\exp\left(0.421-0.0251\times PC1+0.0178\times PC2+0.0105\times PC3\right)}$$
where *PC*1, *PC*2, and *PC*3 correspond to the first, second, and third PCA components, respectively, as described above. Our classifier was trained on the training data 20 times in total; each time, the MDDI’s ability to distinguish depressed from non-depressed subjects was quantified by ROC analysis. The best cut-off value indicated by each curve was recorded, and eventually, averaged across the 20 trials. The confusion matrix in Table 5 summarizes the diagnostic performance of this aggregated cut-off value on the training set; it achieved excellent diagnostic performance (sensitivity: 0.935, specificity: 0.896, accuracy: 0.911). For comparison purposes, the ROC curve in Figure 1 displays the classifier’s performance on the training set over a range of MDDI values (AUC: 0.97). When the classifier was run using the best cut-off value indicated by this curve, its performance was excellent and comparable to that achieved by the mean cut-off value (sensitivity: 0.948, specificity: 0.896, accuracy: 0.917).

### 3.4. Model Testing

The classification performance of the MDDI cut-off value derived above in distinguishing depressed from normal subjects was tested on the test dataset. Table 6 shows the resulting confusion matrix (sensitivity: 0.800, specificity: 0.786, accuracy: 0.795).

### 3.5. Effects of Recording Environment (Facility) and HDRS Score

Figure 2 shows the distributions of MDDI among subjects recruited from each facility. “H1dep” and “H1nor” denote depressed and normal subjects recruited from H1, respectively. Normal subjects’ mean MDDI was compared between facilities using the Steel–Dwass method for distribution-free multiple comparisons [34]. The differences between the following pairs were statistically significant: H1nor vs. U2 (p=0.00123**), H2 vs. U1 (p=0.0186*), and H2 vs. U2 (p=0.0000133**). The differences between the following pairs were non-significant: H1nor vs. H2 (p=0.999), H1nor vs. U1 (p=0.0866), and U1 vs. U2 (p=0.727).

Depressed subjects’ mean MDDI was compared using Welch’s *t*-test (assuming their distributions had unequal variance). The difference between C and Hdep was statistically significant (t(40.06)=2.49,p=0.0170*). Since these subgroups also had dissimilar distributions of HDRS score, we conjectured that the observed difference in MDDI was attributable to an underlying difference in depression severity between patients at the two facilities, and tested our hypothesis using the analysis of covariance (ANCOVA) with HDRS score as the covariate. Figure 3 presents scatterplots of HDRS score versus MDDI for depressed patients at each of the two facilities. Regression lines (predicting MDDI) are indicated in red. First, the interaction between facility and HDRS score-reflected by the degree of parallelism between the red lines in Figure 3a,b was not significant (p=0.577), thereby confirming homogeneous correlation between the covariate and MDDI at each facility (a necessary assumption of ANCOVA). Next, the residuals between observed and predicted MDDI from the covariate were compared between facilities using ANOVA, and the difference was not significant (p=0.159).

The correlation strength between MDDI and HDRS score was 0.186 at C (p=0.105), 0.278 at H1dep (p=0.178), and 0.285 overall (p=0.00374**).

### 3.6. Age Differences

The age distribution in the depressed group and healthy groups differ. As seen in Table 1, almost no elderly subjects are present in the depressed group. Since voice quality differs between the young and elderly age groups, it may affect classification. We therefore conducted a covariance analysis with age as a covariate, similar to Section 3.5. Figure 4a shows the age distribution for the depressed and healthy groups, and Figure 4b shows the MDDI and age distribution. The blue and green lines in Figure 4b represent the regression lines for the MDDI of the depressed and healthy group, respectively. No significant interaction was observed between age of the depressed group and age of the healthy group (p=0.334). Accordingly, the correlation between the covariate and MDDI is consistent for all groups. A significant difference ($p<0.01^{**}$) could be confirmed when comparing the residuals between the observed and predicted MDDI of the groups using variance analysis.

### 3.7. Gender Differences

Figure 5 shows the gender-specific distributions of MDDI among subjects recruited from each facility. Within each facility, the mean HDDI was compared between men and women using Welch’s *t*-test. Significant gender-related differences were not observed (C: t(47.03)=−0.37,p=0.716, H1dep: t(6.91)=1.24,p=0.255, H1nor: t(8.46)=−1.04, p=0.327, H2: t(18.69)=0.20,p=0.843, U1: t(20.21)=0.28,p=0.779, U2: t(47.29)=0.73, p=0.469).

## 4. Discussion

Early in the model development, the fact that nearly 200 features computed by openSMILE exceeded our high cut-off value for feature selection (AUC > 0.85) led us to expect an abundance of major differences in vocal acoustic qualities of depressed patients compared with those of normal adults. However, the fact that over 80% of their variance could be explained by just three principal components suggested that these differences could be captured by a limited set of qualities. Nevertheless, the sheer number of features with strong loadings on each component made it difficult to interpret what specific vocal properties each represented. The nature of vocal properties influenced by major depressive disorder still remains a mystery. Audio features need to be mapped to physiological attributes of voice in order to decipher the meaning of these components. Hence, it remains an option for future work. The marginally less than 200 openSMILE features selected did not include any features relating to F0 and MFCC2, demonstrated by Mundt et al. [18] and Taguchi et al. [26] to be effective in distinguishing patients with major depressive disorders. One of the reasons behind this is believed to be the difference in the voice format, since Mundt et al. analyzed telephone speech and Taguchi et al. analyzed speech in a 16-bit, 22.05kHz PCM format. In addition, MFCC is a feature that highly depends on the content of speech, and differences in the content of speech may also have an effect.

The MDDI is calculated using logistic regression with regularization of three components derived by PCA. This criterion demonstrated very good classification performance (AUC > 0.95), distinguishing between depressed and normal subjects with sensitivity, specificity, and accuracy close to 0.9. This finding supports our expectation of major differences in vocal qualities between depressed and non-depressed adults, and suggests that such differences were properly captured by our classification criterion. Good performance was also observed when the MDDI was used to classify subjects in the test dataset—with sensitivity, specificity, and accuracy values close to 0.8—thus confirming our algorithm did not overfit to the training data. However, further validation seems necessary because we excluded patients that were considered to be in remission (HDRS score < 8), meaning that the sample size of the test set was not fully preserved.

Despite normalizing audio files before analysis to minimize the effects of recording environment, some indications suggested that we might have been unable to eliminate sources of variability besides volume. For normal adults, significant differences in mean MDDI were observed between recordings made in hospital examination rooms and university lecture halls; however, facility-related differences were not observed between comparable environments (i.e., H1nor/H2 and U1/U2). Since none of the depressed patients were recorded in a lecture hall, it is unclear how environmental differences could affect the proposed model’s ability to distinguish them from controls. In addition, it is noteworthy that samples taken from conference rooms tended to have lower MDDI than those from examination room recordings; if the same tendency occurred among patients, it could compromise our model’s detection performance. On the other hand, the fact that depressed patients recorded at C had significantly higher mean MDDI than those at H1dep was not attributable to environmental differences per se; instead, it could be explained by the fact that depression was originally more severe (i.e., HDRS score was higher) at patients at C than H1dep. Indeed, the difference in mean MDDI disappeared after adjustment for HDRS score; this supports the conclusion that facility-related differences in MDDI actually originate from facility-related differences in HDRS score. Furthermore, MDDI appears to reflect depression severity; this hypothesis is supported by the fact that it did not significantly correlate with HDRS score at any individual facility, but did correlate—albeit weakly—across the entire sample.

Almost no elderly patients were included in the depressed group, while some elderly patients were included in the healthy group. Since voice quality generally changes with age, it may have contributed to the classification. We therefore adjusted the MDDI for the two groups by age. Subsequently, a comparison of the mean values showed significant difference. Hence, we can conclude that differences in age distribution have no statistical impact on the MDDI. The reason for this is believed to be that subjects of all ages were included in the healthy group, and any features correlating to age were eliminated in the training process.

Hormonal changes affect the voice characteristics [35] and impact the MDDI discrimination. Therefore, this phenomenon needs to be considered. Women are more susceptible to hormonal changes than men, owing to the menstruation. Therefore, we compared the gender differences in the MDDI values. We could not confirm statistical differences in MDDI among depressed or control subjects within any participating facility. The extraction of features with a significant male-female difference from the selected openSMILE features resulted in 43 and 59 features for the depressed and healthy group, respectively. After excluding similar feature pairs from each group, we had 8 and 10 features for the depressed and healthy group, respectively. Accordingly, the absence of any male–female difference in the MDDI may be due to the absence of varying features between males and females in the MDDI. The hormonal changes during the menstruation affect shimmering and jittering speech features [36]. We did not detect statistically gender-based hormonal differences, as the MDDI excluded the shimmering and jittering speech features. A slight dissimilarity is visible between MDDI distributions of men and women within H1dep and H1nor. While the shimmering and jittering speech features remain unaltered, several other features were affected by the hormonal changes. Although gender-based hormonal changes need to be considered, minimal gender-based differences were observed in the MDDI. Since our classifier is gender-neutral, it is unnecessary to switch models according to patient gender, and thus seems easier to implement compared to the methodology proposed by Jiang et al. [23]. The sound of the voice impacted hormonal changes [37]. However, such correlation was not relevant to our MDDI discrimination.

Differences in experimental conditions have a vital impact on the accuracy of research in this field. Therefore, subject conditions (age, gender, recording environment, etc.) must be as consistent as possible. However, since the health status of healthy subjects is based on self-reporting, verifying its reliability is difficult. Moreover, defective speech recording may occur due to errors committed by the recording equipment operator, or the comorbidities of depression may be missed due to failure of diagnosis by the physician. Accordingly, a limitation of this study is the difficulty in completely eliminating these factors.

## 5. Conclusions

This study aimed to develop a composite index based on vocal acoustic features that can accurately distinguish patients with major depressive disorder from healthy adults. The data were split into a training set and test set in advance. The voice data in the training set were processed using openSMILE to derive 6552 vocal acoustic features. To prevent overfitting, the full set of features was screened to select those that seemed most useful for classification. Next, dimensionality was further reduced by combining and transforming qualitatively similar features using PCA. Logistic regression with regularization was then applied using the three resulting components as model parameters. The proposed criterion—MDDI—distinguished between depressed and normal subjects with ~90% sensitivity, specificity, and accuracy in the training set, and ~80% sensitivity, specificity, and accuracy in the test set. The near absence of gender-related differences in MDDI provides further support of its potential efficacy in practice.

Still, several topics require further study. Clarification is needed on the nature of vocal properties affected by depression. Differences in recording environments, such as between examination rooms in hospitals and conference rooms in general use buildings, should be eliminated as much as possible. Finally, the MDDI’s diagnostic performance should be tested on larger samples of test data.

## Figures and Tables

**Figure 1 ijerph-19-11397-f001:**
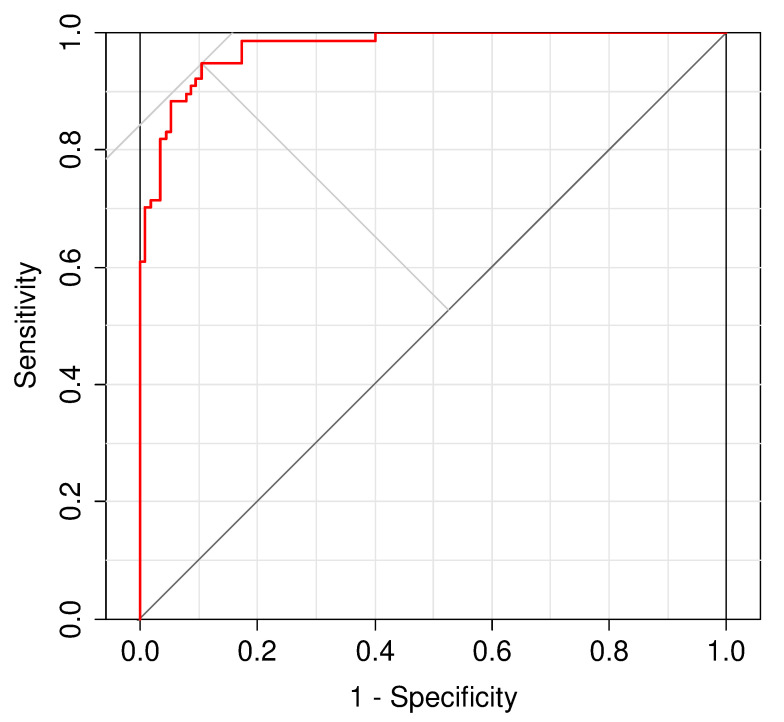
Classification performance of MDDI on training data (ROC curve).

**Figure 2 ijerph-19-11397-f002:**
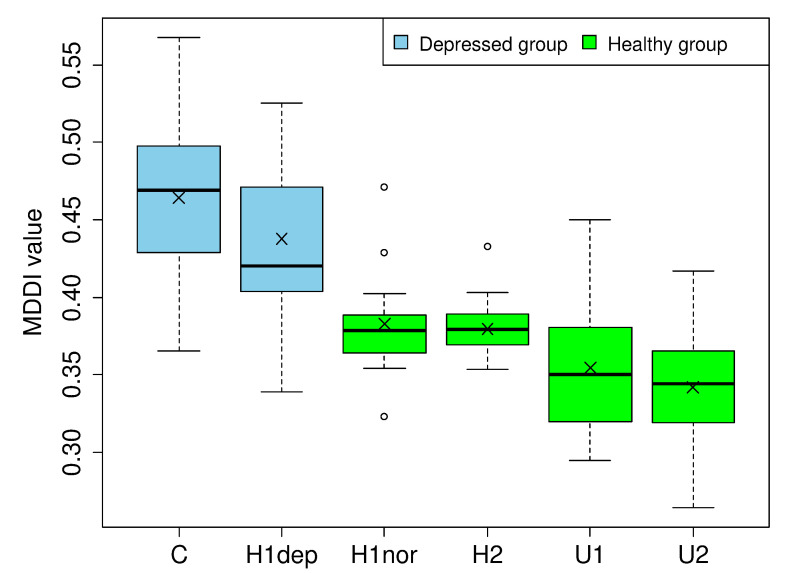
MDDI distributions by facility.

**Figure 3 ijerph-19-11397-f003:**
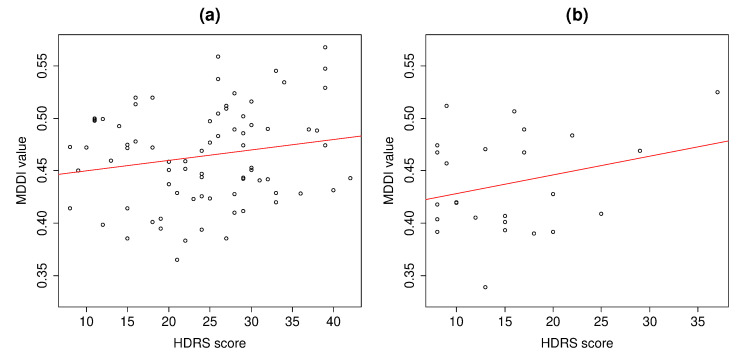
Scatterplots of MDDI versus HDRS score of depressed subjects by facility: (**a**) C; (**b**) H1 (H1dep).

**Figure 4 ijerph-19-11397-f004:**
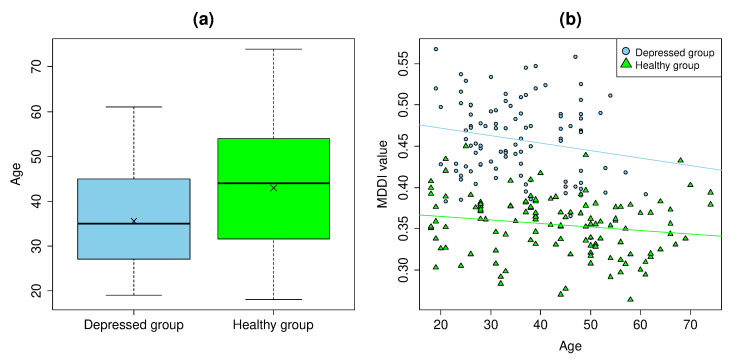
(**a**) Age distribution and (**b**) scatterplot of MDDI versus age for the depressed and healthy groups.

**Figure 5 ijerph-19-11397-f005:**
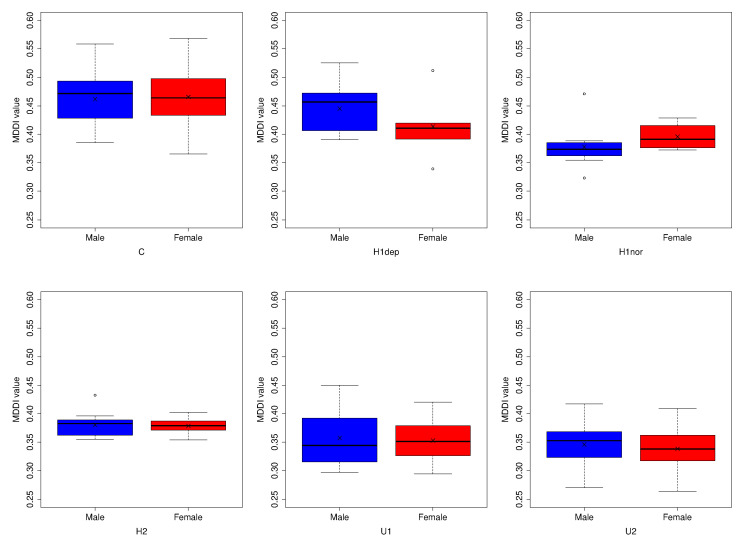
MDDI distributions by facility (gender-specific).

**Table 1 ijerph-19-11397-t001:** Patient information by facility.

Facility	Gender	Number of Depressed Subjects	Number of Healthy Subjects	Age (Mean ± SD)
C	Male	32	0	32.7 ± 6.6
	Female	55	0	31.6 ± 8.6
	Total	87	0	32.0 ± 7.9
H1	Male	46	10	48.1 ± 12.9
	Female	44	4	61.2 ± 13.9
	Total	90	14	54.1 ± 14.8
H2	Male	0	12	47.6 ± 10.6
	Female	0	11	59.8 ± 13.3
	Total	0	23	53.4 ± 13.2
U1	Male	0	14	37.1 ± 18.0
	Female	0	24	37.3 ± 17.8
	Total	0	38	37.2 ± 17.6
U2	Male	0	25	37.9 ± 8.1
	Female	0	29	47.0 ± 11.7
	Total	0	54	42.8 ± 11.1

**Table 2 ijerph-19-11397-t002:** Clinical information of MDD patients. Remission group is patients with HDRS < 8. Depression group is patients with HDRS ≥ 8. The M and F symbols in number of subjects mean male and female, respectively.

Facility	Group	Number of Subjects	HDRS (Mean ± SD)	Analysis
C	Remission	10 (M: 7, F: 3)	4.8 ± 1.4	Not used
	Depression	77 (M: 25, F: 52)	24.3 ± 8.6	Used
	Total	87 (M: 32, F: 55)	22.1 ± 10.2	-
H1	Remission	65 (M: 27, F: 38)	2.2 ± 2.2	Not used
	Depression	25 (M: 11, F: 14)	15.3 ± 7.3	Used
	Total	90 (M: 38, F: 52)	5.8 ± 7.3	-

**Table 3 ijerph-19-11397-t003:** Set phrases read aloud by subjects.

Phrase Number	Japanese Phrase	English Translation
P1	Totemo genki desu	I am very cheerful
P2	Kinō wa yoku nemuremashita	I slept well yesterday
P3	Shokuyoku ga arimasu	I have an appetite
P4	Kokoro ga odayaka desu	My heart is calm
P5	Tsukarete guttari shiteimasu	I’m dead tired
P6	Okorippoi desu	I am irritable
P7	I–ro–ha–ni–ho–he–to	(No meaning; like “a–b–c”)
P8	Honjitsu wa seiten nari	“It is fine today” (standard radio test)
P9	Mukashi mukashi aru tokoro ni	“Once upon a time, there lived…”
P10	Garapagosu shotō	Galapagos Islands

**Table 4 ijerph-19-11397-t004:** Regression coefficients (mean).

	Regression Coefficient
Intercept	−0.421
Principal component 1	0.0251
Principal component 2	−0.0178
Principal component 3	−0.0105

**Table 5 ijerph-19-11397-t005:** Classification performance of MDDI on training data (confusion matrix).

		Predicted Group	
		Depressed Group	Healthy Group
Actual group	Depressed group	72	5
	Healthy group	12	103

**Table 6 ijerph-19-11397-t006:** Classification performance of MDDI cut-off on test data (confusion matrix).

		Predicted Group	
		Depressed Group	Healthy Group
Actual group	Depressed group	20	5
	Healthy group	3	11

## Data Availability

The data presented in this study are available on request from the corresponding author. The data are not publicly available due to privacy and ethical restrictions.

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
