# Peer review of "Detection of Major Depressive Disorder Based on a Combination of Voice Features: An Exploratory Approach"

_ijerph, 2022, doi:10.3390/ijerph191811397_

Round 1

Reviewer 1 Report

The task of discriminating major depressive disorder based on a prosody compelling in itself. The strength of this study is a competent statistical approach to identifying the most informative signs of prosodic control (MDDI).

However, to understand the merits of the study, it is necessary to describe the design of the experiment more clearly from a neurobiological point of view and make comparisons taking into account the neurobiological factors that determine prosodic control.

Comments for the Statement 

a. Considering that MDD state of affairs requires technologies that are capable of easily checking for mental illnesses, for which early intervention is associated with higher remission rates This claim needs more valid confirming reference than just Okuda, A. et al., 2010. Look 47 results in PubMed data base https://pubmed.ncbi.nlm.nih.gov/?term=EEG+predictors+of+MDD&size=20

Comments for the Methods

a. It’s too complicated understand the experimental design and the rationale of separating the group according 

b. Why were the processed data split by facility of origin into a training set (C, 137H2, U1, U2) and test set (H1)? 

c. Table 1 demonstrates invalid comparison these different aged groups that should be reflected in prosody control. (Participants of H1 and H2 are older than C and U1) It would be clearer if authors join depressive patients in age groups separately from healthy subjects.

e. Moreover, it is not reasonable to determine gender specificity because it could be expecting that sound of a voice is responsible for the hormonal differences (Seltzer et al.,. Evol Hum Behav. 2012 Jan;33(1):42-45. doi: 10.1016/j.evolhumbehav.2011.05.004.), that could be observed in women in different hormonal states connected with menstrual cycle emotional voice fluctuations (https://pubmed.ncbi.nlm.nih.gov/? erm=Prosody++menstrual&size=20)

Author Response

Point 1: Considering that MDD state of affairs requires technologies that are capable of easily checking for mental illnesses, for which early intervention is associated with higher remission rates This claim needs more valid confirming reference than just Okuda, A. et al., 2010.

Response 1: Thank you for your comments. We have added new references [5] and [6].

Point 2: It’s too complicated understand the experimental design and the rationale of separating the group according. Why were the processed data split by facility of origin into a training set (C, 137H2, U1, U2) and test set (H1)?

Response 2: Thank you for your comments. We have described this point at p.5, l.161-166.

Point 3: Table 1 demonstrates invalid comparison these different aged groups that should be reflected in prosody control. (Participants of H1 and H2 are older than C and U1) It would be clearer if authors join depressive patients in age groups separately from healthy subjects.

Response 3: Thank you for your comments. We have included additional analysis at p.8, l.222-234 and Figure 4. We have also discussed this point at p.10, l.290-297.

Point 4: Moreover, it is not reasonable to determine gender specificity because it could be expecting that sound of a voice is responsible for the hormonal differences (Seltzer et al.,. Evol Hum Behav. 2012 Jan;33(1):42-45. doi: 10.1016/j.evolhumbehav.2011.05.004.), that could be observed in women in different hormonal states connected with menstrual cycle emotional voice fluctuations.

Response 4: Thank you for your comments. We have discussed this point at p.11, l.299-304, l.306-307, and l.310-311. We think Seltzer's paper you presented is different from the intent of your comment. The paper focuses on girls 7.5-12 years old, but our study subjects are boys and girls 18 years old and older. Also, this paper does not analyze any changes in the voice itself. So we think this paper is not relevant to our study. However, gender differences are clear in voice, so we have added another references [35].

Reviewer 2 Report

Congratulations on your approach in detecting major depressive disorder based on a combination of voice features: an exploratory approach. That said, it would have been great if the introduction to the paragraph had explained the correlation between depression and voice trait.

The methods and methods section is written correctly, the results are presented clearly.

I would suggest improving the discussion section and clearly presenting the limitations of the study and discussing our own results with those of other authors.

I appreciate the idea of the paper and look forward to a minor correction.

Author Response

Point1: it would have been great if the introduction to the paragraph had explained the correlation between depression and voice trait.

Response 1: Thank you for your comments. We have described this point at p.2, l.53-54.

Point2: I would suggest improving the discussion section and clearly presenting the limitations of the study and discussing our own results with those of other authors.

Response 2: Thank you for your comments. We have described the limitation of our study at p.11, l.312-318. Moreover, we have discussed our results with those of other authors at p.10, l.253-259.

Reviewer 3 Report

The paper proposed detection of major depressive disorder based on a combination of voice features: an exploratory approach. However, the authors should mention data split for training and testing, What type of classification is used for final result as well as data distribution in Table 2 confusion matrix should be double checked.

Author Response

Point 1: the authors should mention data split for training and testing.

Response 1: Thank you for your comments. We have described this point at p.5, l.161-166.

Point 2: What type of classification is used for final result as well as data distribution in Table 2 confusion matrix should be double checked.

Response 2: Thank you for your comments. We have added a column to Table 2 indicating which data were used in the analysis. There are 177 depressed patients in total, but because we excluded patients with HDRS less than 8, 102 patients were used in the analysis. The confusion matrices are correct because there are 77 depressed patients in the training data and 25 depressed patients in the test data.

Round 2

Reviewer 1 Report

I have to repeat  very obvious claim that sound of a voice is responsible for the hormonal differences that is observed in women of reproductive age  in different hormonal states connected with menstrual cycle emotional voice fluctuation (even in girls at 12 years! Seltzer et al.,. Evol Hum Behav. 2012 Jan;33(1):42-45. doi: 10.1016/j.evolhumbehav.2011.05.004.) I suggest authors should take in account this natural phenomen.

Author Response

Point 1: I have to repeat  very obvious claim that sound of a voice is responsible for the hormonal differences that is observed in women of reproductive age in different hormonal states connected with menstrual cycle emotional voice fluctuation (even in girls at 12 years! Seltzer et al.,. Evol Hum Behav. 2012 Jan;33(1):42-45. doi: 10.1016/j.evolhumbehav.2011.05.004.)

I suggest authors should take in account this natural phenomenon.

Response 1: We acknowledge the reviewer for this comment. However, we would like to rebut the claim that “the impact of the sound of the voice on hormonal changes should be considered” in the section on gender differences, as follows.

“The sound of the voice impacts hormonal changes [37]. However, such correlation was not relevant to our MDDI discrimination.” We aimed to detect the stress variations using the vocal sounds instead of the hormones. When the subject’s stress and hormonal levels altered upon listening to a familiar person’s voice, the subject’s voice reflected the stress variations. Furthermore, the impact of the sound of the voice on hormones was not limited to women. Seltzer’s study appeared to be based on the journal’s theme, human evolution. As all the subjects in this study were female, the impact of the sound of the voice on hormones could not be ascertained as a women-specific response. The same phenomenon is likely to occur in men. As the cortisol and oxytocin are gender-independent hormones, the impact of the sound of the voice on hormones should not be considered in the gender difference sections.

The hormonal changes affect the vocal sounds and consequently impact our observations.  Therefore, we discussed these variations. Women are more susceptible to hormonal changes than men, owing to the menstruation. Therefore, we compared the gender differences in the MDDI values. We did not observe statistically significant gender differences. Hormonal changes during the menstruation affect the shimmering and jittering speech features [1]. Therefore, we did not detect statistically significant gender-based hormonal differences, as our dataset excluded the shimmering and jittering speech features. The shimmers and jitters are not the only speech features that are affected by hormones. Therefore, hormonal changes need to be considered.

The above concepts are discussed in p. 12, I. 299-303, 309-311, 312-315, and 318-319 of the manuscript.

[1] Chae SW, Choi G, Kang HJ, Choi JO, Jin SM. Clinical analysis of voice change as a parameter of premenstrual syndrome. J Voice. 2001 Jun;15(2):278-83. doi: 10.1016/S0892-1997(01)00028-5. PMID: 11411481.